# Construction of Microporous Zincophilic Interface for Stable Zn Anode

**DOI:** 10.3390/molecules28124789

**Published:** 2023-06-15

**Authors:** Xin Yang, Tie Shu, Haoyu Huang, Hongquan Yi, Yanchi Zhang, Wei Xiao, Liang Li, Yuxin Zhang, Minghao Ma, Xingyuan Liu, Kexin Yao

**Affiliations:** 1School of Chemistry and Chemical Engineering, Chongqing University, Chongqing 400044, China; xyang0610@163.com (X.Y.); shutie950112@163.com (T.S.); 2Undergraduate School, Chongqing University, Chongqing 400044, China; 17815041238@163.com (H.H.); 19132050489@163.com (H.Y.); 15772046781@163.com (Y.Z.); 13368183178@163.com (W.X.); 3Department of Sciences and Engineering, Sorbonne University Abu Dhabi, Abu Dhabi P.O. Box 38044, United Arab Emirates; liang.li@sorbonne.ae; 4College of Material Science and Engineering, Chongqing University, Chongqing 400044, China; zhangyuxin@cqu.edu.cn; 5Hang Tian School Affiliated to HSXJTU, Xi’an 710043, China; minghaomacqu@163.com; 6Chongqing Joint School of Famous Schools, Chongqing 400030, China; wenqin@cqu.edu.cn; 7State Key Laboratory of Coal Mine Disaster Dynamics and Control, Multi-Scale Porous Materials Center, Institute of Advanced Interdisciplinary Studies, School of Chemistry and Chemical Engineering, Chongqing University, Chongqing 400044, China

**Keywords:** aqueous zinc ion batteries, MOFs, anode, dendrite growth

## Abstract

Aqueous zinc ion batteries (AZIBs) are promising electrochemical energy storage devices due to their high theoretical specific capacity, low cost, and environmental friendliness. However, uncontrolled dendrite growth poses a serious threat to the reversibility of Zn plating/stripping, which impacts the stability of batteries. Therefore, controlling the disordered dendrite growth remains a considerable challenge in the development of AZIBs. Herein, a ZIF-8-derived ZnO/C/N composite (ZOCC) interface layer was constructed on the surface of the Zn anode. The homogeneous distribution of zincophilic ZnO and the N element in the ZOCC facilitates directional Zn deposition on the (002) crystal plane. Moreover, the conductive skeleton with a microporous structure accelerates Zn^2+^ transport kinetics, resulting in a reduction in polarization. As a result, the stability and electrochemical properties of AZIBs are improved. Specifically, the ZOCC@Zn symmetric cell sustains over 1150 h at 0.5 mA cm^−2^ with 0.25 mA h cm^−2^, while the ZOCC@Zn half-cell achieves an outstanding Coulombic efficiency of 99.79% over 2000 cycles. This work provides a simple and effective strategy for improving the lifespan of AZIBs.

## 1. Introduction

The excessive consumption of non-renewable energy sources has caused global warming, frequent disasters, and energy shortage, all of which seriously affect the environment. To mitigate these issues, renewable energy sources have gradually emerged into human view [1]. Although lithium-ion batteries (LIBs) have been widely utilized in electronic products and new energy vehicles, they still face some problems, such as limited resources, high costs, and safety hazards [2,3,4,5]. Aqueous zinc ion batteries (AZIBs) have captured the attention of researchers due to their inherent characteristics, including low cost, high safety, abundant resources, high theoretical capacity (820 mA h g^−1^), and low redox potential (−0.76 V) [6,7,8,9],. However, a series of issues such as dendrite growth, dead Zn, and by-product generation severely constrains the practical application of AZIBs.

To address the issues mentioned above, researchers have proposed various methods including interfacial modulation [10,11,12,13,14], electrolyte optimization [15,16,17], and electrode structure design [18,19,20]. Interfacial modulation is a promising approach to control the electrode surface reaction, which effectively suppresses dendrite formation and regulates Zn deposition by manipulating the interface structure and composition. Previous studies have demonstrated that a microporous structure not only increases the Zn^2+^ conductivity and adjusts the interfacial electric field to achieve uniform Zn^2+^ flow, but also modifies the coordination of [Zn(H_2_O)_6_]^2+^ during deposition. For instance, microporous materials were utilized as an artificial interface to promote uniform Zn deposition [21]. Limiting the electrolyte in nanometer-sized pores prevented interface side reactions and dendrite penetration [22]. MCM-41 mesoporous molecular sieves were employed to construct dendrite-free anodes [23]. In addition, zincophilic factors regulate the deposition’s behavior and improve the anode’s lifespan. Zhou et al. [24] designed a nitrogen-doped graphene oxide (NGO) composite interface layer with zincophilic nitrogen functional groups that strongly adsorb Zn atoms, directing their deposition on the (002) crystal plane and suppressing hydrogen evolution and passivation. Ma et al. [25] constructed a ZnO interface layer in situ on the Zn anode, which provides a desolvation barrier and excellent zincophilic ability. Despite the fact that progress in optimizing the electrochemical performance has been made, the issues of dendrite growth and extending the cycling life of AZIBs remain a tremendous challenge.

Metal-organic frameworks (MOFs) and their derivatives have gained remarkable attention as potential materials for energy storage due to their controllable composition, pore size, and high specific surface area. Supercapacitors [26], LIBs [27], and fuel cells [28] are the fields where MOFs and its derivatives have been utilized. Fu et al. [29] fabricated a porous Mn-based cathode material with N-doping (MnO_x_@N-C) via a MOFs template strategy. The MnO_x_@N-C cathode exhibits favorable cycling stability and high reversibility, owing to its conductivity and porous structure. The cathode consistently maintained a capacity of 100 mA h g^−1^ during 1600 cycles at a high rate of 2000 mA g^−1^, which was superior to most reported ZIB cathode materials. Moreover, MOFs and derivatives can be modified to generate numerous zincophilic sites while retaining their original structure, which is a natural advantage in enhancing the stability of AZIBs.

In this study, we obtained the ZOCC material by thermally treating the pre-synthesized cubic structure of the ZIF-8 precursor at a high temperature in a nitrogen atmosphere. Thermal treatment not only allows ZOCC to inherit the skeleton structure of ZIF-8, but also transforms Zn^2+^ to ZnO in situ, which results in uniform distribution of ZnO in the porous skeleton structure. Because the ZOCC skeleton is loaded with zincophilic ZnO and the N element, the Zn^2+^ transporting to the interface is adsorbed onto ZnO and the N element with zincophilic traits, inducing the directional deposition of Zn at (002) a crystal plane and reducing dendrite formation, thereby prolonging the cycling life. Additionally, faster kinetic transport and lower polarization potential are delivered in the ZOCC@Zn anode, because of the unique porous and conductive carbon structure. Concretely, at the condition of 0.5 mA cm^−2^ and 0.25 mA h cm^−2^, the symmetric cell sustains 1152 h. Furthermore, the ZOCC@Zn anode achieves a remarkable CE of 99.79% (2016 cycles) at a current density of 5 mA cm^−2^ with a capacity of 1 mA h cm^−2^, demonstrating its outstanding electrochemical performance. For ZOCC@Zn//MnO_2_, the specific capacity is 128.8 mA h g^−1^ with a capacity retention of 67.51% after 400 cycles, which is superior to the bare Zn full cell. Overall, our study provides a novel perspective for designing dendrite-free, low-polarization potential anodes for AZIBs.

## 2. Results and Discussion

Figure 1a presents a schematic diagram of the preparation process of ZOCC@Zn. Porous ZOCC was obtained by carbonizing the synthesized cubic ZIF-8 precursors, and ZOCC@Zn was further obtained by uniformly coating ZOCC on bare Zn. The X-ray diffraction (XRD) pattern of the synthesized ZIF-8 (Appendix A) matches well with the typical diffraction peak of the simulated card [30], indicating a successful synthesis with excellent crystallinity. After calcination at 800 °C in a nitrogen atmosphere, the diffraction peak of ZIF-8 completely disappeared. The XRD pattern of the ZOOC (Figure 1b) shows two distinct diffraction peaks with broad half-peaks at 24° and 43°, corresponding to the (002) and (100) crystal planes, respectively. The weak peak of the (100) belongs to amorphous carbon, and the signal on the (002) crystal plane is graphitized carbon. No diffraction peak related to Zn is observed. This absence may be due to the content of the Zn element evaporating to a very low level under a high temperature condition. We specifically discuss the existence form of the Zn element in the subsequent XPS analysis. Additionally, the specific surface area and pore size distribution of the ZOCC were investigated using N_2_ adsorption–desorption isotherms. Figure 1c displays the sharp increase at a relatively low pressure (P/P_0_ ≤ 0.01), exhibiting the microporous structure [31]. The appearance of a hysteresis loop indicates the presence of mesopores. The Brunauer–Emmett–Teller (BET) surface area is 675.72 m^2^ g^−1^ with a total pore volume of 0.591 cm^3^ g^−1^. The hierarchical pore structure of ZOCC with a high specific surface area facilitates the kinetic mass transfer and regulates the equilibrium Zn^2+^ ion flux [32]. Furthermore, the pore size distribution curve (Figure 1d) was obtained by using the non-local density functional theory (NLDFT) method. The pore size distribution of ZOCC is mainly in the range of 0.6–4 nm, revealing that ZOCC is predominantly microporous (<2 nm) with a slight mesoporous pore structure (2–50 nm). X-ray photoelectron spectroscopy (XPS) was employed to analyze the composition and chemical bonding state of ZOCC. As shown in Appendix A, the wide-scan XPS spectrum clearly demonstrates the presence of C, N, and Zn elements. The high-resolution XPS spectrum of C 1s (Figure 1e) can be deconvoluted into three peaks, which are ascribed to C sp^3^-C sp^3^ (284.8 eV), C sp^2^-C sp^2^ (286.6 eV), and C=N bonds (289.7 eV) [33,34]. The presence of the C=N bonds indicates that the carbon skeleton of ZOCC is doped with the N element, which is consistent with the high-resolution XPS spectrum of N 1s. As displayed in Figure 1f, the peaks at 398.4, 399.7, and 401.0 eV are assigned to pyridinic-N, pyrrolic-N, and graphitic-N, respectively. It is demonstrated that most N atoms are integrated into the carbon lattice and the doping of N atoms is essential to enhance the electrical conductivity of the carbon material and the electrochemical properties. Furthermore, the presence of pyridinic-N and pyrrolic-N enhances the wettability of the composite surface, thereby facilitating the adsorption of Zn^2+^ and promoting electron transport and zinc ion diffusion [32,35]. The Zn 2p spectrum (Figure 1g) indicates binding energies of 1021.6 eV and 1044.6 eV for Zn 2p_3/2_ and Zn 2p_1/2_, respectively. The 23.0 eV binding energy gap between the two peaks confirms the presence of Zn^2+^ originating from ZnO [36,37].

The morphology and structure of both ZIF-8 and ZOCC powder were characterized by scanning electron microscopy (SEM) and transmission electron microscopy (TEM). The obtained ZIF-8 precursor (Figure 2a,b) exhibits a cubic shape with a smooth surface and an average size of about 200 nm, which is consistent with the TEM image displayed in Appendix A. The uniform distribution of C, N, O, and Zn elements in ZIF-8 was confirmed by the corresponding elemental mapping. After calcination, as shown in Figure 2c,d, ZOCC generally inherits the original shape of the precursor, but with a surface decorated with granular substances, which are identified as ZnO by the elemental analysis of HAADF-STEM in Figure 3. This observation is consistent with the XPS results. Although there is some degree of shrinkage and diameter reduction in ZOCC compared to the precursor, the structure remains intact without any visible collapse. This shrinkage may be attributed to the volatilization of Zn or the depletion of C and N under a high calcination temperature.

We fabricated the ZOCC interface layer on the bare Zn surface by mixing ZOCC powder and a small amount of organic binder PVDF to prepare a slurry. Appendix A displays the overall flatness of the bare Zn surface with a thickness of 80 μm (Appendix A). However, at a higher magnification, the image in Appendix A reveals some defects on the surface. Under the effect of organic binder, ZOCC is evenly distributed on the electrode surface (Appendix A) without altering its original morphology (Appendix A). Furthermore, the cross-sectional SEM image confirms that the ZOCC layer is densely spread on the Zn metal surface with a thickness of about 20 μm (Appendix A). To assess the electrochemical stability of ZOCC@Zn, symmetric cells were assembled using 2 M ZnSO_4_ solution as the electrolyte. The ZOCC@Zn and bare Zn symmetric cells separately underwent cyclic plating/stripping tests at different current densities and capacities. As depicted in Figure 4a, the cycling life of ZOCC@Zn can stably reach 1152 h at a current density of 0.5 mA cm^−2^ and a capacity of 0.25 mA h cm^−2^, while maintaining a continuous ultralow polarization (18 mV). Conversely, the bare Zn cell displays clear potential fluctuations, a high polarization of approximately 48 mV, and a short cycling life of 165 h. As the current density is increased to 1 mA cm^−2^ and the capacity to 1 mA h cm^−2^ (Figure 4b), the ZOCC@Zn cell still achieves a relatively long cycling time of 519 h with a lower polarization (19 mV). In sharp contrast, the bare Zn cell exhibits a higher polarization of about 37 mV. What is more, at a current density of 2 mA cm^−2^ (Figure 4c), the polarization potential of bare Zn gradually declines, with a sudden drop at 228 h, presumably due to the uncontrolled growth of dendrites caused by inhomogeneous deposition until the diaphragm is punctured and the cell is short-circuited. However, the ZOCC@Zn cell can be plated/stripped for 720 h. Finally, further galvanostatic charge/discharge testing was performed at 5 mA cm^−2^ as shown in Figure 4d, demonstrating that the ZOCC@Zn cell has a lower polarization and longer cycle number than bare Zn. These findings suggest that ZOCC@Zn is better suited for long-term cycling at various current densities and is adaptable to more diverse environments, while bare Zn is prone to short-circuiting. It also well demonstrates that the ZOCC interface layer effectively inhibits dendrite growth and prolongs battery life.

ZOCC@Zn and bare Zn symmetric cells were serially measured at current densities (1, 2, 5, 8, and 10 mA cm^−2^) with a fixed areal capacity of 1 mA h cm^−2^ to evaluate the stability and rate aspects (Figure 4e). The polarization of the ZOCC@Zn cell is consistently lower than that of the bare Zn cell at every current density, demonstrating that the ZOCC@Zn cell has excellent Zn^2+^ transport kinetics together with plating/stripping stability. In addition, we examined the effect of the ZOCC layer on the reaction kinetics by obtaining the relationship between the exchange current density and the Zn deposition kinetics based on the following equation: [38,39]
$$i=i_{0}\frac{F\eta_{total}}{2RT}$$
where i, $i_{0}$, and $\eta_{total}$ represent the cycling current density, exchange current density, and total overpotential, respectively. *F*, *R*, and *T* refer to the faradic constant, the ideal gas constant, and the Kelvin temperature, respectively. The calculations in Appendix A reveal that the exchange current density of the ZOCC@Zn anode (9.12 mA cm^−2^) is superior to that of the bare Zn cell (7.05 mA cm^−2^), which means that the kinetics of ZOCC@Zn cell is promoted.

The morphological characterization of the electrode surface after 15 cycles was performed to further validate the modulation effect of the ZOCC interface layer. Appendix A shows that Zn^2+^ prefers to deposit more uniformly on the ZOCC@Zn surface compared to bare Zn. In addition, the distinctive peak at 8° of ZnSO_4_(OH)_6_·5H_2_O (PDF#39-0688) is clearly observed in the XRD pattern of the bare Zn electrode surface after cycling, which is not found on the ZOCC@Zn surface (Figure 4f). When the current density and capacity are increased to 1, the surface of the bare Zn electrode has an irregular and disordered morphology with a large number of protrusions, as shown in Appendix A. However, the ZOCC@Zn surface exhibits a relatively flat and dendrite-free morphology, with the uniformly deposited Zn forming a hexagonal lamellar structure (Appendix A). Furthermore, at 2 mA cm^−2^, 1 mA h cm^−2^, uneven deposition and severe dendrite growth appeared in bare Zn, resulting in clear regional overgrowth and continuous accumulation on the electrode, which is extremely detrimental to cell reversibility (Figure 5a,b), In contrast, the surface of the ZOCC@Zn electrode is apparently flat and the metal Zn is uniformly distributed on the (002) crystal plane, as shown in Figure 5c,d. Finally, ZOCC@Zn at 5 mA cm^−2^, 1 mA h cm^−2^ still had a remarkable effect in restraining dendrite growth (Appendix A).

Coulombic efficiency (CE) is another critical parameter for evaluating electrochemical performance. A half-cell was assembled with ZOCC@Zn or bare Zn as the anode and copper foil as the cathode for charge/discharge tests. For bare Zn//Cu, a severe fluctuation after 81 h was observed at 2 mA cm^−2^, 1 mA h cm^−2^. Encouragingly, the ZOCC@Zn//Cu presents an exceptional average CE of 99.60% for 808 h (Figure 6a). The charge/discharge voltage profiles of ZOCC@Zn//Cu and bare Zn//Cu are exhibited in Figure 6b and Figure 6c, respectively. It can be clearly seen that ZOCC@Zn//Cu has great reversibility in the Zn plating/stripping process. Notably, when the current density is increased to 5 mA cm^−2^ (Figure 6d), the ZOCC@Zn//Cu displays an outstanding average CE (99.79%) for 2016 cycles, which is remarkably superior to that of the bare Zn//Cu (252 cycles). Moreover, the voltage stability of ZOCC@Zn//Cu is significantly stronger than that of bare Zn//Cu (Figure 6e,f), reflecting the fact that the reversibility of the electrode is remarkably raised by constructing the ZOCC interfacial layer.

To explore the practical feasibility of the ZOCC@Zn anode, bare Zn//MnO_2_, and ZOCC@Zn//MnO_2_, full cells were assembled to evaluate the contribution of the ZOCC layer in improving the performance of AZIBs. α-MnO_2_ was synthesized based on previous literature [15]. The excellent crystallinity of α-MnO_2_ was confirmed by XRD (Appendix A). SEM (Appendix A) and TEM (Appendix A) were employed to confirm the nanorod-like structure of α-MnO_2_. Cyclic voltammetry (CV) curves were initially measured to understand the electrochemical behavior of the cathode and anode, as shown in Figure 6g. Typical MnO_2_ redox peaks are clearly observed in ZOCC@Zn and bare Zn full cells, corresponding to the reversible redox reaction between MnO_2_ and MnOOH [40]. This denotes the fact that both full cells have similar electrochemical behavior and the ZOCC coating does not impair the redox process of MnO_2_. Interestingly, the gap between the redox peaks of ZOCC@Zn//MnO_2_ is reduced compared to bare Zn//MnO_2_, which represents a decrease in voltage hysteresis. Moreover, ZOCC@Zn//MnO_2_ has a higher current density, illustrating a faster plating/stripping rate and a higher capacity. The electrochemical impedance (EIS) studies also proved these results. Figure 6h was obtained by fitting the Nyquist plot to the equivalent circuit of Appendix A. The EIS curve distinctly reveals a lower charge transfer resistance (R_ct_) of 109.4 Ω for the ZOCC@Zn//MnO_2_, compared to 351.2 Ω for bare Zn//MnO_2_, which illustrates that the presence of the ZOCC layer decreases the interfacial resistance, rendering the charge transfer and kinetics of Zn^2+^ faster.

Figure 6i confirms the capacity reversibility of the ZOCC@Zn cell, which varies smoothly with current density and shows a higher specific capacity at each current density than bare Zn//MnO_2_. In addition, the cycling performances of full cells at 1 A g^−1^ are shown in Figure 6j; the specific capacity of ZOCC@Zn//MnO_2_ is 128.8 mA h g^−1^ with a capacity retention of 67.51% after 400 cycles while maintaining an admirable average CE (99.72%). However, for bare Zn//MnO_2_, the capacity dropped extremely fast, leaving only 34.71% of its initial discharge specific capacity after 400 cycles. The corresponding voltage profiles are shown in Figure 6k,l. The capacity of the bare Zn full cell drops rapidly from the initial 222.8 mA h to 86 mA h after 200 cycles. Unusually, for the ZOCC@Zn cell, the capacity only varies from the initial 229 mA h to 200.6 mA h. This phenomenon is mainly attributed to the flat deposition morphology of the ZOCC@Zn cell, which limits the generation of dead Zn during cycling. In light of the above results, the approach of the ZOCC interfacial modification is very promising for promoting the practical application of AZIBs.

## 3. Materials and Methods

### 3.1. Materials

The materials required, namely zinc nitrate hexahydrate (Zn(NO_3_)_2_·6H_2_O), 2-methylimidazole (2-MI), hexadecyltrimethylammonium bromide (CTAB), zinc sulfate heptahydrate (Zn(SO_4_)_2_·7H_2_O), potassium permanganate (KMnO_4_), manganese acetate tetrahydrate (Mn(CH_3_COO)_2_·4H_2_O), manganese sulfate monohydrate (MnSO_4_·H_2_O), polyvinylidene fluoride (PVDF), acetylene black, and 1-Methyl-2-pyrrolidinone (NMP), were all obtained from the Shanghai Aladdin Biochemical Technology Co., Ltd. (Shanghai, China). All reagents were analytical grade and used without any further purification.

### 3.2. Preparation of ZIF-8 Precursor and ZOCC

A modified method was used to synthesize the ZIF-8 at room temperature [41]. Solution A was prepared by completely dissolving 0.713 g of Zn(NO_3_)_2_·6H_2_O in 25 mL of deionized water and dissolving 12 mg of CTAB and 10.9 g of 2-MI in 160 mL of deionized water with ultrasonic stirring for 30 min to prepare Solution B. Subsequently, Solution A was gradually added to Solution B and stirred for 15 min at room temperature. The white product was collected by centrifugation and washed with deionized water and ethanol, respectively, and then dried under vacuum at 70 °C for 12 h to obtain the ZIF-8 precursor. The obtained ZIF-8 powder was placed in a tubular furnace, raised to 800 °C with a heating rate of 5 °C min^−1^ and maintained for 2 h in the N_2_ atmosphere. The calcined powder was collected and denoted as ZOCC.

### 3.3. Preparation of ZOCC@Zn

To fabricate the ZOCC@Zn, a homogeneous slurry was prepared by mixing the as-prepared ZOCC powder and PVDF in the mass ratio of 9:1, and grinding it with NMP solvent. The slurry was then uniformly distributed on the Zn foil (80 μm) and dried at 60 °C for 12 h to obtain ZOCC@Zn.

### 3.4. Preparation of MnO_2_ Electrode

MnO_2_ powder was prepared based on previous reports [15,42]. Solutions C and D were attained by dissolving 4.74 g of KMnO_4_ and 11.03 g of Mn(CH_3_COO)_2_·4H_2_O in 40 mL of deionized water, respectively. Solution D was gradually added to solution C while stirring vigorously, and heated for 4 h at 80 °C in a water bath. The precipitate was separated by centrifugation and washed with deionized water, then dried at 80 °C for 12 h. Subsequently, the precipitate was heated to 200 °C in a tube furnace with air to obtain the desired MnO_2_ powder.

The MnO_2_ electrode was fabricated by grinding MnO_2_ powder, acetylene black, and PVDF into a slurry in the weight ratio of 7:2:1 with the NMP solvent. The slurry was coated on Ti foil (20 μm) and dried under vacuum at 70 °C for 12 h. The loading amount of active substance MnO_2_ was about 1.2 mg.

### 3.5. Characterization

The microscopic morphology and particle size of the samples were characterized using scanning electron microscopy (SEM, Helios 5 CX, Thermo Scientific, Waltham, MA, USA) and transmission electron microscopy (TEM, Talos, F200S, Thermo Scientific, Waltham, MA, USA G2). The crystal structure and phase composition were analyzed using X-ray diffraction (XRD, Empyrean, Panalytical B.V., Almelo, The Netherlands) with Cu Kα radiation between 5° and 80° (40 kV; 40 mA; 5° min^−1^). The specific surface area and pore size distribution can be obtained from the adsorption−desorption isotherm of nitrogen at −196 °C (Belsorp Max, MicrotracBEL, Osaka, Japan); the sample should be degassed at 200 °C for 12 h before testing. Additionally, X-ray photoelectron spectroscopy (XPS, K-Alpha, Thermo Scientific, Waltham, MA, USA) was obtained using a thermo scientific Kα energy spectrometer paired with an X-ray source of monochromatic Al-K_α_.

The electrochemical properties were tested mainly for symmetric cells, half-cells, and full cells. For symmetric cells, ZOCC@Zn served as the working electrode and counter-electrode. The half-cell was assembled with ZOCC@Zn as the negative electrode and Cu foil (20 μm) as the positive electrode. Furthermore, with the MnO_2_ electrode as the positive electrode, the full cell was assembled. A series of electrochemical tests was performed with bare Zn instead of ZOCC@Zn electrodes as a comparison while maintaining all test conditions. The above electrodes were sized as 12 mm diameter rounds and assembled into CR-2032-type coin cells at an air atmosphere with a glass fiber (0.68 mm, Whatman, GF/D) as the separator, where the electrolyte for symmetric and half-cells was 90 mL of 2 M Zn(SO_4_)_2_ and for the full cells was 90 mL of 2 M ZnSO_4_ + 0.2 M MnSO_4_. A Land CT3002A test system (Wuhan LAND Electronic Co. Ltd., Wuhan, China) was employed for galvanostatic measurements of coin cells at different current densities of 0.5–5 mA cm^−2^ and capacities of 0.25–1 mA h cm^−2^. Cyclic voltammetry (CV) and electrochemical impedance (EIS) tests were carried on a CHI660E electrochemical workstation.

## 4. Conclusions

In summary, ZOCC with a microporous framework structure was applied to the surface modification of a Zn anode as a way to optimize the electrochemical properties of AZIBs. Initially, the conductive ZOCC interface layer with a microporous structure could homogenize the electric field distribution on the electrode surface, thereby improving the reversibility of the plating/stripping process, accelerating the Zn^2+^ transport kinetics, and reducing the polarization potential. Furthermore, ZnO and N atoms with zincophilic trait were uniformly distributed in the ZOCC framework, which attracted the directional Zn deposition, thus effectively suppressing the formation of dendrites during charging/discharging. Notably, the unique structure and composition of the ZOCC layer contributed to addressing dendrite growth and achieving excellent electrochemical performance. In detail, the ZOCC@Zn symmetric cell at a current density of 0.5 mA cm^−2^ and a capacity of 0.25 mA h cm^−2^ can cycle stably for 1152 h with an ultralow polarization of 18 mV. Meanwhile, at 2 mA cm^−2^, 1 mA h cm^−2^, an outstanding average CE (99.79%) for over 2016 cycles was observed in the ZOCC@Zn//Cu. The capacity retention of the ZOCC@Zn//MnO_2_ was greater than that of the bare Zn cell after 400 cycles. This work offers a novel insight into the inhibition of dendrite growth and accelerating the evolution of AZIBs in practical applications.

## Figures and Tables

**Figure 1 molecules-28-04789-f001:**
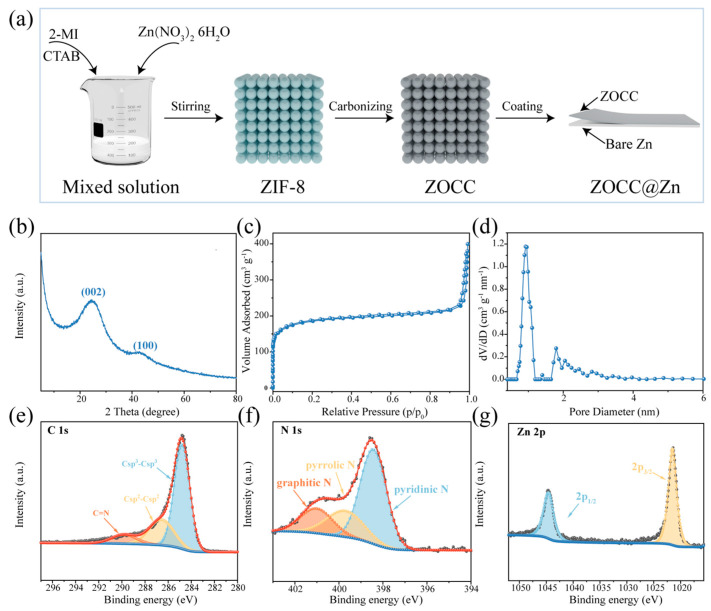
(**a**) Schematic illustration of the preparation process of the ZOCC@Zn; (**b**) XRD pattern of ZOCC; (**c**) N_2_ adsorption/desorption isotherm of ZOCC and (**d**) Corresponding pore size distribution; XPS spectra of ZOCC: (**e**) C 1s; (**f**) N 1s; (**g**) Zn 2p.

**Figure 2 molecules-28-04789-f002:**
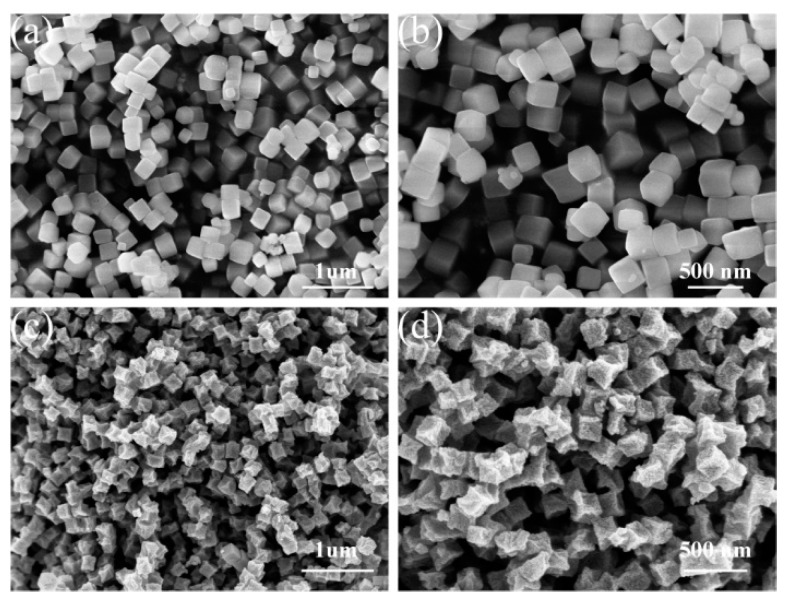
SEM patterns of (**a**,**b**) ZIF-8 and (**c**,**d**) ZOCC.

**Figure 3 molecules-28-04789-f003:**
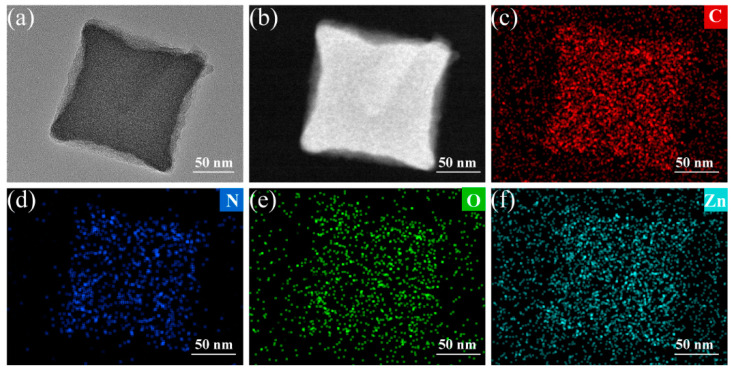
(**a**) TEM image of ZOCC; (**b**) HAADF-STEM image of ZOCC; Corresponding elemental mapping of (**c**) C; (**d**) N; (**e**) O; (**f**) Zn.

**Figure 4 molecules-28-04789-f004:**
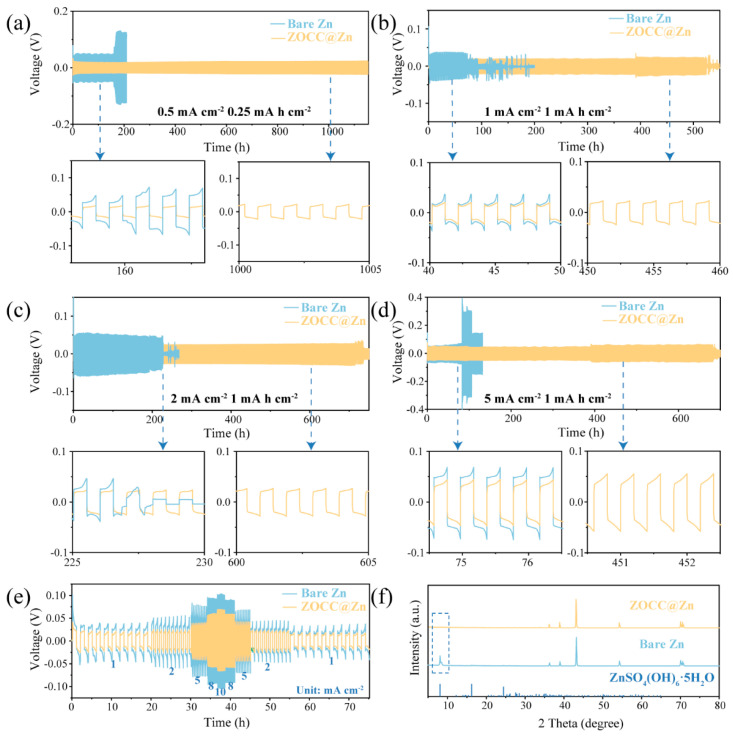
(**a**–**d**) Long-term galvanostatic cycling of bare Zn and ZOCC@Zn symmetric cells at different current densities and capacities; (**e**) Rate performance of bare Zn and ZOCC@Zn symmetric cells; (**f**) XRD patterns of Zn metal surfaces in bare Zn and ZOCC@Zn symmetric cells after 15 cycles at a current density of 0.5 mA cm^−2^ and a capacity of 0.25 mA h cm^−2^.

**Figure 5 molecules-28-04789-f005:**
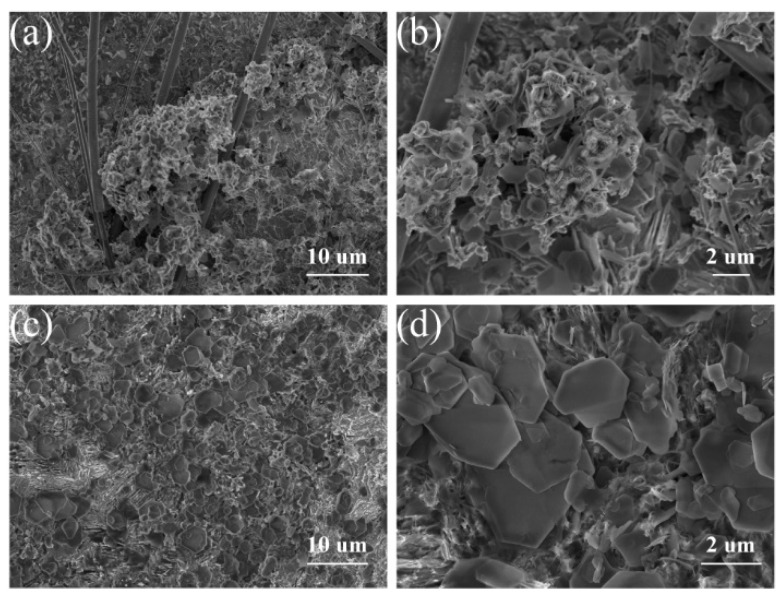
Surface morphology of bare Zn (**a**,**b**) and ZOCC@Zn (**c**,**d**) electrodes after 15 cycles at a current density of 2 mA cm^−2^ and a capacity of 1 mA h cm^−2^, respectively.

**Figure 6 molecules-28-04789-f006:**
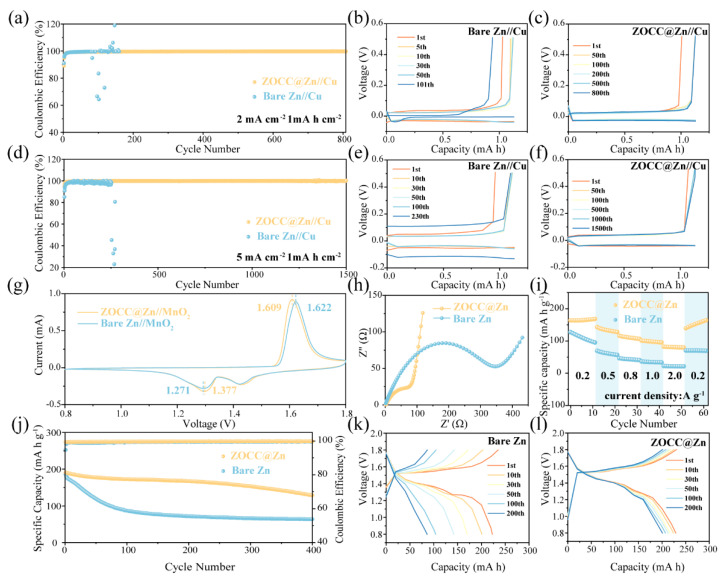
CE comparison of bare Zn//Cu and ZOCC@Zn//Cu at the current densities of 2 mA cm^−2^ (**a**) and 5 mA cm^−2^ (**d**) with a capacity of 1 mA h cm^−2^; the corresponding charge/discharge voltage profiles of bare Zn//Cu (**b**,**e**) and ZOCC@Zn//Cu (**c**,**f**) with different cycle number, respectively; (**g**) CV curves of ZOCC@Zn//MnO_2_ and bare Zn//MnO_2_ within a voltage window of 0.8–1.8 V with a scan rate of 0.1 mV s^−1^; (**h**) Nyquist plots of ZOCC@Zn//MnO_2_ and bare Zn//MnO_2_; (**i**) Rate capability of ZOCC@Zn//MnO_2_ and bare Zn//MnO_2_ cells at various current densities from 0.2 to 2.0 A g^−1^; (**j**) Long-term cycling performance of ZOCC@Zn//MnO_2_ and bare Zn//MnO_2_ full cells at 1 A g^−1^; Voltage profiles of bare Zn//MnO_2_ (**k**) and ZOCC@Zn//MnO_2_ (**l**).

## Data Availability

Data of the compounds are available from the authors.

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
