# Peer review of "Construction of Microporous Zincophilic Interface for Stable Zn Anode"

_molecules, 2023, doi:10.3390/molecules28124789_

Round 1

Reviewer 1 Report

This paper "Construction of microporous zincophilic interface for stable Zn anode" presented a method for interfacial modulation of ZIF-8-derived ZOCC material used as anodes. A series of comparative electrochemical performance tests were performed and the results were analyzed and explained. The article exhibited the novelty of the results and the rigor of the logic. The authors demonstrated that the prepared material is promising as interfacial layer for optimizing electrochemical properties. However, several aspects are recommended to improve and polish.

Comment 1: Please check the manuscript carefully for type and formatting mistakes. For example, in line 109, the “70 ℃” shows different types.

Comment 2: What are the advantages of ZOCC obtained by high-temperature thermal treatment of ZIF-8 in a nitrogen atmosphere? Why was it chosen as the material for interface modulation?

Comment 3: The experimental section should be more detail, the specifications of the Cu and Ti foils used for the tests, and the amount of electrolyte should be stated clearly.

Need to improve the English. 

Reviewer 2 Report

Xin Yang and co-workers report the fabrication and characterization of ZnO/C/N composite (ZOCC), a derivative of zeolitic imidazolate frameworks (ZIF-8), and claim the superior properties of ZOCC as an anode of electrochemical storage devices. The result shows that better performance of ZOCC than that of Zn, and it may attract the community of aqueous zinc ion batteries research. However, I think, the most important information which is the characterization and identification of synthesized ZOCC is not clear. It is fatal as the report on the newly developed materials. Although the authors show some characterization data on ZOCC, such as X-ray diffraction (XRD), X-ray photoelectron spectroscopy (XPS), scanning electron microscopy (SEM), and transmission electron microscopy (TEM), ZOCC was not identified. The authors should clearly show what is the ZOCC. Other concerns (not all) are listed below.

1)     The description of “ZOCC@Zn” is quite confusing. ZOCC is not included in Zn. It should be “ZOCC on Zn”.

2)     As far as I understand, “Solutions A and B” in section 2.4 are different from “Solutions A and B” in section 2.2. If so, they should be “Solutions C and D”.

3)     In line of 148, the authors show that the diameter of the anode was 20 MICRO meters in the 2023 cell. I feel it is quite small from the standard experimental viewpoint. Is it correct?

4)     The authors claimed that the sudden degradation of the Zn electrode cell originated from the short-circuit by the dendrites of Zn. Although a dendrite-like structure was observed, is there any evidence of short-circuit by the dendrites?

From the above reasons, the authors are advised to revise the manuscript to satisfy the standard criteria as a journal paper, which shows the basic identification of products.

Some grammatical errors are found.

Round 2

Reviewer 1 Report

Accept.

It is acceptable for MDPI standards. 
